# Advanced Sentiment Analysis for Managing and Improving Patient Experience: Application for General Practitioner (GP) Classification in Northamptonshire

**DOI:** 10.3390/ijerph20126119

**Published:** 2023-06-13

**Authors:** Aavash Raj Pandey, Mahdi Seify, Udoka Okonta, Amin Hosseinian-Far

**Affiliations:** Department of Business Systems & Operations, University of Northampton, Northampton NN1 5 PH, UK; aavash.pandey18@my.northampton.ac.uk (A.R.P.); mahdi.seify@northampton.ac.uk (M.S.);

**Keywords:** classification, sentiment analysis, National Health Service, bot, patient experience

## Abstract

This paper presents a novel analytical approach for improving patients’ experience in healthcare settings. The analytical tool uses a classifier and a recommend management approach to facilitate decision making in a timely manner. The designed methodology comprises of 4 key stages, which include developing a bot to scrap web data while performing sentiment analysis and extracting keywords from National Health Service (NHS) rate and review webpages, building a classifier with Waikato Environment for Knowledge Analysis (WEKA), analyzing speech with Python, and using Microsoft Excel for analysis. In the selected context, a total of 178 reviews were extracted from General Practitioners (GP) websites within Northamptonshire County, UK. Accordingly, 4764 keywords such as “kind”, “exactly”, “discharged”, “long waits”, “impolite staff”, “worse”, “problem”, “happy”, “late” and “excellent” were selected. In addition, 178 reviews were analyzed to highlight trends and patterns. The classifier model grouped GPs into gold, silver, and bronze categories. The outlined analytical approach complements the current patient feedback analysis approaches by GPs. This paper solely relied upon the feedback available on the NHS’ rate and review webpages. The contribution of the paper is to highlight the integration of easily available tools to perform higher level of analysis that provides understanding about patients’ experience. The context and tools used in this study for ranking services within the healthcare domain is novel in nature, since it involves extracting useful insights from the provided feedback.

## 1. Introduction

Patient feedback is a terminology used to describe feedback received from patients that are in various forms of experience, satisfaction, assessments, and viewpoints within the boundaries of quality, continuity, and accessibility. With the integration of the internet into users’ daily lives and the adoption of social media platforms, feedback has become more of a social activity [1]. Timely and unsolicited feedback with a focus on the true voice of users creates a dynamic feedback process [2]. Although various forms of patient feedback practices exist, their use is found to be limited to hospital administration and management contexts [3]. Accordingly, sentiment analysis is effective in providing insights into how strongly opinionated the patient feedback is, whilst providing the healthcare providers with the customer’s opinion about their product or service [4,5]. There are currently a wide range of sentiment analysis studies that are focused on the synthesis of data from websites and various social media platforms such as Twitter, Facebook, etc. [5,6]. Such data could entail positive, negative, or neutral subjective opinions and feelings, which could be expressed towards a specific area of interest [7,8]

Developing an effective communication channel is usually the first step for involving patients in this process. Gathering feedback provides the substantial information required to drive the continuous improvement of internal processes and can accordingly inform strategy to drive change. Feedback can be received via numerous communication channels such as phone calls, text messages, online reviews, customer service desk feedback forms, surveys, questionnaires, etc, even though, there are limitations with all of these channels; institutions, however, can provide innovative solutions that are precise in nature to overcome the such issues [9]. With the fear that direct patient feedback to general practitioners (GPs) might break patient confidentiality, the use of third-party services via the United Kingdom’s National Health Service (NHS) effectively meets this need [10]. Thus, the NHS has provided a rate and review section on its website to allow its userbase to offer feedback about the services received. The NHS believes that with this approach, GPs will have the opportunity to get feedback on their services and provide a direction on where improvement should be directed [11]. However, a labeling system that ranks the best performing GP within a user’s proximity, thus creating a benchmark, is not available on the NHS rate and review website. This is due to various reasons, including ensuring that each GP has sufficient patients to cater for.

A GP patient survey within Northamptonshire identified various reasons that support the ranking of GPs, including ease of access, booking an appointment with a doctor, and attentiveness of the receptionists and healthcare staff to patients, etc. [12]. This survey is important as the demand for healthcare services has been on the rise, while the number of GPs has not grown to meet the demand. According to the NHS, each GP in Northamptonshire had an average of 119 more patients between 2020 and 2021 [13]. Improving GP services has become a challenge for the NHS and a suitable management approach is required to mitigate this crisis. A recent study shows that 42% of internet users in the UK have access to provide some form of online feedback, yet only 8% had given feedback regarding their healthcare experience [14]. This could be because patients will only give feedback if their responses will have an impact [15].

This paper aims to outline a novel monitoring and evaluation system, with a view to tackling shortcomings with the current rate and review system, as outlined above. The paper is organized as follows: Section 2 provides a critical review of the existing literature around patients’ feedback, and also a concise overview of machine learning and natural language processing techniques used within the work. Section 3 outlines the study methodology, followed by the results in Section 4. Finally, Section 5 and Section 6 entail a set of recommendations and a summary of the key findings, respectively.

## 2. Literature Review

According to Santana [16], health care systems can only improve efficiently if the system is developed in line with the patient centered care (PCC) model. One of the aims of PCC is to get patients’ feedback to improve their experience [17] by embracing the verbal, the non-verbal, and the entire patient management experience. Feedback is expected to be specific, void of emotional blackmail, timely, and credible [10]. However, due to the complexity and interlinkage of departments within healthcare systems, applying feedback might be challenging, as there is usually a mistrust of the data, and the system is not trained to adopt agile continuous improvements [18]. Thus, in order for the feedback to be effective, it must be easy to analyze and prioritize variables, whilst implementation is coordinated by a total quality management (TQM) team [19].

This study builds on (Figure 1) a framework developed by Sheard et al. [18] which combines three concepts used to drive change in the healthcare sector. The first concept is normative legitimacy (NL) which looks at what the healthcare institution believes is the right thing to do. The second concept is structural legitimacy (SL) which looks at the powerplay, hierarchy, and entrepreneurial mindset within the institution. Finally, organizational readiness (OR) focuses on the willpower to create and implement change across the institution. The analysis, validation, and prioritization of feedback must serve as a basis for OR and should be guided by the institution’s strategic objective and resource capacity.

A fourth layer called monitoring and evaluation (M&E) is introduced which ensures there is a regular review of planned changes and implementation from the board level. This framework then has an overarching and iterative continuous improvement (CI) built within its system to make “improvement” a way of life rather than a project. The introduction of CI would give the customer-facing staff a holistic view of how feedback is used [15]. Across NL, SL, and OR, it is obvious that all structures have been put in place to receive patient feedback, making it a standard requirement. However, how the feedback is utilized differs across the board. In the review of normative legitimacy, feedback is assigned to specific healthcare staff within the GP practice who decide on the legitimacy of the feedback based on the value placed on it. The structural legitimacy is affected by multi-tiered issues which could range from hierarchical powerplay in being able to implement feedback void of bureaucracy, accepting ownership and assigning responsibility, and having the resources and capacity to review and implement patient feedback. Accordingly, a few questions arise: Are there laid down process rules that help determine who can resolve specific patient feedback? Can this process flow be regularly updated to ensure all staff can see their level of responsibility for specific feedback without it being burdensome on the already strained system? What happens when there are resource constraints?

At the level of organizational readiness, the following questions are raised: Is the system fully developed with the capacity and resources at both the departmental and institutional level to drive change? Does communication and collaboration within the institution flow like the Kanban process, ensuring that there is always a seamless handshake and a positive outlook by all departments on system improvement? Implementation of feedback at this level would mean that feedback has been properly analyzed, validated by the assigned owners, and prioritized in line with the institution’s objective and growth strategy.

Monitoring and evaluation has been set up as a standard process across all structures, as the board sets a feasible quality benchmark and provides the resources to meet the benchmarks. Continuous improvement must be embedded within the system wherein changes and learning points from previous feedback implementation are proactively reviewed. Therefore, the implementation team identifies its role in the change, the challenges faced, and how those challenges were dealt with. Using this approach, the implementation team can propose better operational approaches for handling such changes in a way that there is an appropriate documenting process flow which can be easily applied in subsequent requests.

### 2.1. Feedback Practices among GPs

Technological advancements have made it possible for institutions to gather information in numerous ways. The methods of digital data extraction will continuously evolve and will likely be able to generate even more data in the future. Digital solutions such as Augmented Reality (AR), IoT, and Artificial Intelligence (AI) embedded within digital and social platforms play a crucial role in facilitating the data acquisition process [20]. GPs currently have various techniques for collating patient feedback such as patient panels, patient experience surveys, mystery patient data collection, and focus groups [21]; however, systems and tools should be leveraged to efficiently analyze such big data to generate useful insights. The current practice of feedback-driven improvement has been deemed transformational since it opens a whole new world of communication that the NHS never knew existed [2]; although the possibilities with the use of bots is yet to be imagined. The current patient feedback analysis process has a very slow turnaround time from analysis to impactful change. Thus, a new structure should be designed which is capable of processing real-time or semi-real-time data so that the acquired data can be immediately analyzed, and results can be derived. This study incorporates statistical analysis on the patient feedback data which can be combined with the existing power of BI within the NHS, to perform analysis on the demographic distribution of patients at various GPs. The NHS currently has an interactive dashboard that can be viewed by users for GP appointments, mental health activity, cervical screening, maternity services, among many others [22]. However, patient feedback data are not visible in an interactive and effective manner. The analytical tools currently in use by the GPs have not yet prioritized or placed emphasis on patient feedback. Thus, the feedback process can be improved by leveraging a bot to identify patterns and derive meaningful insights from patient data. Studies also suggest that key areas such as scheduled follow-up, accountability, prioritization of actions, and opportunities should be emphasized to ensure that changes are effectively implemented.

### 2.2. Benchmarking

Benchmarking has been regarded as a mainstream tool for performance management to drive improvement by linking metrics with practice. It allows institutions to meet their targets by evaluating and measuring their performance and process efficiency. However, there is usually no deliberate effort by GPs to surpass benchmarks and raise the quality bar [23]. Benchmarking has been divided into four distinct types: Process benchmarking, competitive benchmarking, internal benchmarking, and functional benchmarking [24]. To spur continuous improvement, process benchmarking discovers and implements the best practices by setting a minimum standard for an entire institutional process. In process benchmarking, a crucial step is identifying partners and networks. The partners are the units that deliver information regarding benchmarking investigation whilst the networks are established to coordinate operational activities [25].

### 2.3. Data Analysis

The patient feedback section on the NHS website allows for unstructured and semi-structured data, which leads to the acquisition of huge chunks of data. However, some institutions identify the lack of resources and capacity to review the supplied feedback to provide swift insights for management reviews, despite the availability of the data required. This is because the focus has been placed on gathering data without specifically identifying how this feedback will be analyzed and utilized. Analyzing feedback is an arduous task as it sometimes entails manual review of raw printed documents which can demoralize the reviewers and ultimately make them lose any sense of urgency in going through the reviews [26]. This task is seen as a burden, which is combined with the increased strain on the NHS wherein there are major staff shortages and a rising number of patients [15]. To make patient feedback insightful and easy to review, big data analytics is essential whilst being able to leverage machine learning and natural language processing (NLP) to automate this task and save both the NHS and GPs thousands of pounds and man-hours. The results obtained from machine learning models can be heavily relied upon once a high level of accuracy has been developed. This is expected to significantly improve analysis of the feedback and provide better patient experience going forward, thus showing that the NHS and GPs alike can quickly gain insights from reviews.

Big data refers to a repository or various sources of large chunks of unstructured, semi-structured, and structured data from which rich insights can be developed using the relevant analytical tools [27]. Institutions take advantage of their big data analytics capability to connect users and technology [28] as this has been identified as a cornerstone for competitive success for institutional performance. Adapting the use of big data analytics could serve as a bedrock for structuring and formalizing feedback with the integration of human capital to improve data analytic abilities, structural capital to accelerate decision making, and relational capital to transfer knowledge [28,29]. Electronic medical records support advanced analytics and help to create quick insights on large amounts of data to aid decision making. However, much of the data is usually unstructured, creating a challenge for data grouping and further analysis. Systematic analysis and management have been emphasized to be the crucial factor in harnessing the capabilities of big data [30]. For example, the “All of Us” initiative gathered more than one million health data from patients residing in the US. The aim of the US National Institutes of Health (NIH) is to accelerate research and analyze the data to improve healthcare [31].

#### 2.3.1. Machine Learning

There is a huge demand for automation to effectively manage large amounts of knowledge in big data. This is why machine learning techniques have been considered for its powerful algorithms to acquire knowledge, learn patterns, and make predictions from the data set [32]. Big data classification is not restricted to a single dimension as it has a spectrum with various levels of value that are answered with descriptive, predictive, and prescriptive analytics. Classifier models are adopted and trained using labeled data within select themes during coding, and at each instance, the data is represented with features [33,34]. The principle of machine learning was developed during a computer programming experiment using the rules of checkers, incomplete and redundant parameters, and a sense of direction [35]. The machine would then learn to play it better than the person who programmed it by reviewing patterns and identifying possibilities. Arthur Samuel conducted the above experiment in 1959 and is the first person to propose the concept of machine learning [36]. Machine learning enhances human intelligence in areas dealing with big data. For instance, human intelligence can generate multiple hypotheses while approaching a problem using statistical models and conclude with a false prediction. However, machine learning algorithms reduce the hypothesis by opting to use primary input data and can overcome the shortcomings of bias hypothesis within statistical models [37].

#### 2.3.2. Natural Language Processing (NLP)

NLP uses classification to solve tasks, leveraging computational algorithms to mimic the human language thought process. The approach takes correct parameters while training the classifier so that errors are detected, and the model is improved for higher accuracy [33]. A combination of machine learning algorithms with an NLP classification can be used to solve highly daunting tasks. NLP uses a range of computational techniques for representing and analyzing texts at various levels of linguistic analysis to accomplish human-like language processing. It therefore automates the comprehension and analysis of human languages, enabling users to gain insights with ease [38], and it allows a machine to understand languages used by humans [39]. The text processing level of NLP is at an advanced and critical semantic processing stage, which is a natural ability of humans. There has been an increase in the use of NLP in healthcare for clinical research and quality improvement by extracting valuable information from unstructured data through the system of electronic medical records [40].

#### 2.3.3. Bot

A bot is a software programmed to complete certain tasks in place of humans using the Python programming language to extract data. Upon extraction of reviews, it also performs sentiment analysis from the Valence Aware Dictionary and Sentiment Reasoner (VADER) and extracts keywords from Rapid automatic keyword extraction (RAKE) before storing the results in a CSV format. A bot is developed with packages such as Beautiful Soup, Selenium, VADER, and RAKE. Beautiful Soup is a package in the Python programming language that allows the program to scrape information from the web with ease. The package is preferred ahead of XML or HTML parser as it supplies functions for searching, modifying, and iterating the parse tree. The Beautiful Soup library along with the Selenium WebDriver assists the program to navigate through multiple web pages [41]. RAKE is an algorithm that requires simple input parameters to automatically extract keywords from a body of text by analyzing its coincidence with other collections of words and its frequency count [42]. There have, however, been legal interests surrounding data mining and copyright infringements on data acquisition and usage. Some cases raised have challenged the use of data scraping for research purposes despite being acquired from publicly accessible data [43].

### 2.4. Research Objectives

This study was designed to address the gap in patient experience with GPs. The objectives are as follows:To explore current patient feedback practices within general practitioners (GPs) in Northamptonshire, UK.To recommend ways to improve the patient feedback and associated processes using big data analytics.

### 2.5. Research Questions

The study research questions are as follows:How effective is the patient feedback process in improving healthcare services?How can the feedback process be improved?

### 2.6. Problem Statement

The adoption of tools to perform advanced analysis to improve patient experience facilitates timely and insightful interventions. To improve patients’ experiences, a managerial approach should be practiced that can draw out valuable insights from existing feedback provision systems. Classification techniques that use supervised and unsupervised machine learning techniques can be applied into such a feedback data set. The results of the classification can then be used to group the GPs and highlight areas where GPs are excelling. Managerial approaches such as benchmarking can also be adopted by the NHS to increase patients’ experiences in GPs that have been classified as lower ranked GPs.

## 3. Methods

### 3.1. Data Acquisition

A primary data collection method was used to scrape and load the patient review from the NHS website (accessed on 30 June 2022) into the data model. As reviews are qualitative and difficult to compare, the sentiment analysis with VADER [44] quantifies the review. The results derived from VADER (v. 3.3.2) can then be compared and used for analysis. The keywords extracted from each review are also loaded into a data model. Accordingly:The bot extracts qualitative data from the NHS website and converts it into meaningful quantitative insights.The gathered data are then cleaned, and connections are established using Microsoft Excel.Data modeling is performed with Power Pivot. Pivot tables are created to analyze the data.Data visualization techniques were used to extract insights.For building a classifier model, WEKA (v. 3.9.5) is used as it comprises machine learning algorithms for data mining tasks.

The algorithms within the tool are used to derive advanced analytics and unappreciated information, so that hidden trends can be identified using this approach. The underlying rationale of the analytical tool is to leverage the open source software to perform advanced analytics. Data mining techniques that are powered by machine learning algorithms can be utilized to derive insights from WEKA. The use of the Python programming language enables higher levels of automation as it allows for features that can be performed with voice commands and can be integrated with the interactive analytical tool to provide a userbase with higher levels of visualization.

### 3.2. Ethical Consideration

While web scraping, the researchers operated ethically within the allowed scope by only extracting publicly available data solely for the purpose of this research. Intellectual property was respected and not interfered with, no personally identifiable data was used, and the data used was not shared with third parties.

### 3.3. Experiment Design

VADER: VADER is a tool which is available in the Python library and can be defined as a rule-based algorithm. Moreover, it is a lexicon and rule-based framework for sentiment analysis which was the first choice when compared with seven different sentiment analysis tools [45].

WEKA: Waikato Environment for Knowledge Analysis (WEKA) is a software developed using the Java programming language and is used for data mining [46]; accordingly, it entails a collection of advanced machine learning algorithms. The classifier results and settings have been provided as permitted by WEKA.

## 4. Results

### Classification

The statistical analysis was performed using WEKA, whilst a normalization filter was applied to calculate the threshold value in classifying GPs into three different levels: gold, silver, and bronze. The normalization filter learns properties and patterns of the data set. Then, a classifier model was built with the pre-loaded machine learning algorithms in WEKA. The classifier model requires a labeled data set to determine a particular output. As aggregated reviews were used as a basis for classification, GPs with less than three reviews were neglected. Hence, the classifier model was performed only on 34 GPs within Northamptonshire. The normalization filter derived the threshold scale of the aggregated sentiment value ranging from 0 to 1. An aggregated sentiment value of less than or equal to 0.425 is classified as bronze, a value greater than 0.425 and less than or equal to 0.7 is classified as silver, and, finally, values that are greater than 0.7 are classified as gold. The test mode for the data set is 10-fold cross validation and the findings are depicted below (Figure 2).

In the classification matrix, nine instances of silver, 17 instances of bronze, and five instances of the gold level have been classified correctly. These findings have been highlighted on the main diagonal of the classification matrix. The outcome on the main diagonal indicates correct classification derived from the model. Every outcome outside the main diagonal is a misclassification.

## 5. Discussion

Data mining patient feedback from NHS ratings and reviews provided insights into effective practices that GPs can adopt to improve their services. Although the case study was on GPs based in Northamptonshire County, the analytical tool can be scaled up in terms of operation to include GPs from a wider geographical location across the UK. A recommendation for an additional feature is to analyze data from Twitter, which is also a rich data source, where GPs can analyze the patient feedback posted as tweets. Currently, GPs within Northamptonshire County do not possess an active Twitter account or active websites. Therefore, this feature is recommended for future purposes. Additionally, another recommended feature is to analyze patients’ feedback data from audio sources. With the ability to analyze audio data, programs such as “conversational intelligence” could be designed where an AI program could communicate with a patient, analyze the data, improve the model’s accuracy upon each iteration, and improve its features. The program would update the system and provide its userbase with real-time analysis.

The artifact of the research study complements patient feedback analysis for GPs. More specifically, the research study prioritizes the use of analytical tools to effectively examine relevant data as this will make it easier for analyses. This approach also creates an ultimately improved patient healthcare service experience. Approaches must be updated to leverage technology to create value within the services. Existing relevant literature focuses on hospitals and patient feedback; this study is unique since it concentrates only on the patient feedback practices of GPs. Even though GPs are considered a small proportion of the healthcare system in comparison to hospitals, it is the primary point of contact where a patient makes appointments in relation to their health issue. Sentiment analysis and machine learning techniques can be applied within the data analysis process to gain deeper understanding and discover hidden patterns within the data set. Visualizing the data facilitates the decision-making process. Lastly, the voice assistant feature can also be deployed within the recommended features to provide some level of automation while carrying out data analysis. In summary, this study provides an approach that leverages current software advancement for mining patient feedback data.

The NHS currently focuses on textual data when analyzing patient feedback, however, as a future work the practice of analyzing audio data could enable the NHS to gather feedback from a wider group of patients. From the patient’s perspective, the option of providing feedback with audio data would give them more convenience. In-stead of logging into NHS rate and review website, following the steps for account verification and feedback moderation, the option of verbally providing recent experience would allow more patients to be more expressive about services received. The amount of data acquired with this approach would be tremendous and the data would yield insights that would empower GPs to improve in areas that were previously neglected. Data mining the audio data from patients about their experience would increase the model performance and provide users with real time analysis. The derived insights can then be highlighted to display real time analysis and provide GPs with insights which that can be acted upon. The benefit of using open-source software allows for customization to meet user requirements, so that the analytical tool can be updated to perform sentiment analysis on audio data and linked to display the real time or near to real time data.

## 6. Conclusions

AI programs such as “Conversational Intelligence” can be powered by recommended engines that perform efficiently and with higher accuracy. Machine learning algorithms that learn from data can be applied upon to gain insights from a reliable set of output. The study was centered around text data from the NHS rate and review website. The textual data from patients’ reviews have only been gathered from GPs in the Northamptonshire region, nevertheless the outlined approach can be easily expanded to enable the analyses of feedback on GP clinics across the UK. The findings of the research highlight areas where GPs can improve upon to provide better care for patients. In future, a comparison from public dataset for other GPs within the United Kingdom can provide a much more robust view of the outlined approach. It also worth noting that the choice of dataset size was limited by the actual feedback on the NHS website.

## Figures and Tables

**Figure 1 ijerph-20-06119-f001:**
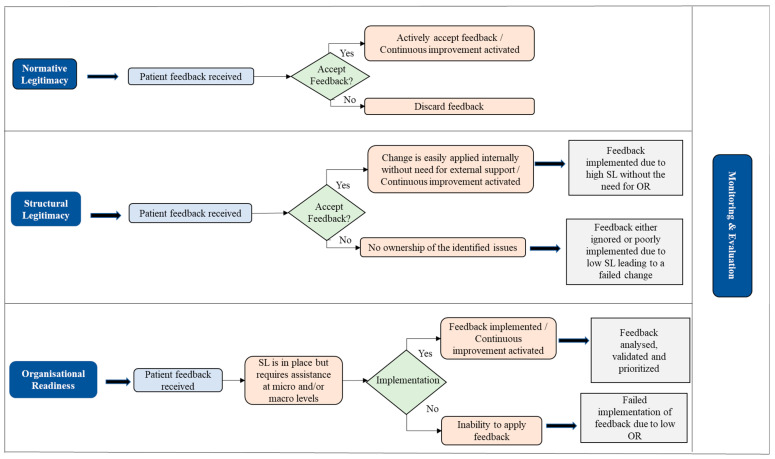
Algorithmic Patient Experience Feedback Framework, adapted from [18].

**Figure 2 ijerph-20-06119-f002:**
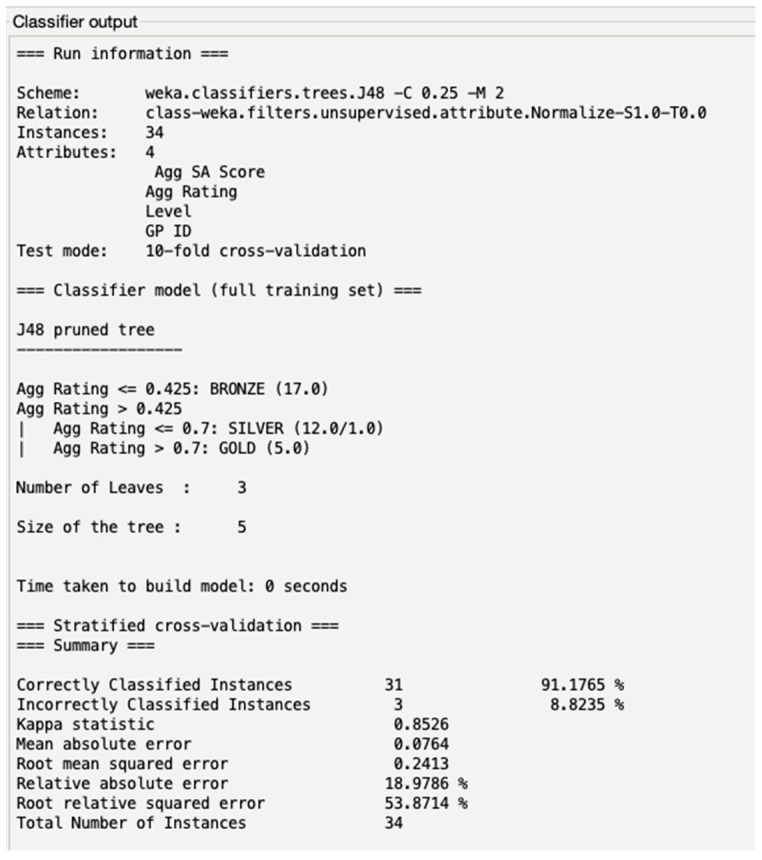
Classification Results for GP Classification.

## Data Availability

Publicly available data sets were analyzed in this study. This data can be found here: [https://www.nhs.uk/service-search/find-a-gp/results/Northampton?latitude=52.23785989953226&longitude=-0.8950457398507264 (accessed on 30 June 2022)].

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
