# Peer review of "Advanced Sentiment Analysis for Managing and Improving Patient Experience: Application for General Practitioner (GP) Classification in Northamptonshire"

_ijerph, 2023, doi:10.3390/ijerph20126119_

Round 1
Reviewer 1 Report
Summary and Overall Contribution:
Sentiment analysis is a crucial tool that can help improve patient experience, and the authors of the paper propose an analytical tool with a machine learning classifier and a recommended management approach to enhance patients' experience in healthcare settings.
Major Comments:
- More information is needed regarding the experimental setting and experimental results, such as the classifier model used.
- A comparison with other public datasets is necessary.
- The dataset size is very small, making it difficult to trust the provided results.
- The related works section is weak concerning works related to sentiment analysis on medical data.
- The authors should better highlight their contribution to justify publication, as they have integrated various basic techniques known and detailed in the literature.
- The authors should compare their work with others published recently.
- The authors focused more on feature work than the actual work, raising questions about the true contribution of the current paper.
- The main contribution of this paper is not clear, as the title, abstract, introduction, and article body seem to tell different stories.
Minor Comments:
- Avoid using words such as "we," "they," and "our," replacing them with the appropriate word.
- There is an ambiguous statement in section 2.3.2: "NLP uses classification to solve tasks, leveraging on computational algorithms to automatically process human languages."
- The authors mentioned a dashboard, which I cannot see in any figure in the paper.
- There is only one figure; more illustrations should be added and described properly.
- The conclusion and future work section is too long and should be revised.
Recommendations:
- Recent deep learning-based methods for sentiment analysis should be used instead of the current methods.
Reviewer 2 Report
Dear authors,I find this study interesting. However, after reading it, some considerations should be made:
Minor issues:
- It is advisable to write the manuscript in an impersonal way. Please avoid using terms such as "we" or "our" throughout the manuscript.
- It is recommended not to include abbreviations in the title of the manuscript.
Major issues:
- In the abstract the authors report this information "In total, 178 reviews were extracted from General Practitioners (GP) websites within Northamptonshire County, 4764 keywords from 178 reviews, and a dashboard was used to highlight trend and patterns". However, I have not been able to find these data throughout the manuscript.
- The manuscript could be improved by following the following structure (Introduction, Methods, Results, Discussion, and Conclusions).
- Both the Methods and Results sections should be more extensively detailed need to be more fully detailed to improve understanding of the process the authors followed and the interpretation of their results.
Regards.
Reviewer 3 Report
This interesting study aimed to introduce the new a novel analytical tool to construct a classifier and a recommend management approach to improve patients’ experience in healthcare settings in the Northamptonshire. The authors designed methodology enables better decision-making relevant manner about GP practice classification in Northamptonshire on the base of patient feedback analysis.
The authors created the classifier model grouped GPs into gold, silver, and bronze categories on the base of sentiment analysis and extracting keywords from National Health Service rate and review webpages, building classifier with Waikato Environment for Knowledge Analysis (WEKA), analysing speech with Python, and using a Microsoft Excel dashboard.
The present study tried to complement the current patient feedback analysis practiced by GPs. The analytical tool facilitates decision-making in a timely and relevant manner about classifier model to group GPs to improve patients’ experience in healthcare settings.
The authors used to collect data in this study 178 reviews were extracted from General Practitioners (GP) websites within the Noramptonshire County. From 4764 keywords from 178 reviews, and a dashboard was used to highlight trend and patterns. I consider the sample size to be sufficient. The analysed group is homogeneous from the point of view of the doctor's place of work. However, we do not have enough information about the respondents.
Were patient reviews anonymous?
Do we know the age structure and gender of the respondents involved in the survey?
Could the results of the study be influenced the factors like age and gender of the respondents?
The measurements and instruments used by the authors seem to be valid. The results are processed in detail and graphically.
The discussion is not part of the paper. It is necessary to add a discussion to the paper for recommendation of the essential findings of the study. More literature can be added to the discussion, enriching the authors' arguments.
The paper does not state the limitations of the results. It is necessary to supplement them. It is not possible to generalize the results of the study, as a specific group of respondents participated in it, about which we do not have enough information.
Why did the authors choose Twitter for data collection?
Why missing another online source of data collection?
Are older respondents also able and interested in using Twitter to provide feedback?
It is necessary to complete the missing parts in the paper. The topic of the paper I evaluate positively due to the presented study finding suggests that data mining patient feedback from NHS ratings and reviews provided insights on effective practices that GPs can adopt to improve their services.
